# Association between Anti-Hepatitis C Viral Intervention Therapy and Risk of Sjögren’s Syndrome: A National Retrospective Analysis

**DOI:** 10.3390/jcm11154259

**Published:** 2022-07-22

**Authors:** Chien-Hsueh Tung, Yen-Chun Chen, Yi-Chun Chen

**Affiliations:** 1Division of Allergy, Immunology and Rheumatology, Department of Internal Medicine, Dalin Tzu Chi Hospital, Buddhist Tzu Chi Medical Foundation, Chiayi 622401, Taiwan; dr5188@yahoo.com.tw; 2School of Medicine, Tzu Chi University, Hualien 97004, Taiwan; bcelltcell@gmail.com; 3Division of Hepato-Gastroenterology, Department of Internal Medicine, Dalin Tzu Chi Hospital, Buddhist Tzu Chi Medical Foundation, Chiayi 622401, Taiwan; 4Division of Nephrology, Department of Internal Medicine, Dalin Tzu Chi Hospital, Buddhist Tzu Chi Medical Foundation, Chiayi 622401, Taiwan

**Keywords:** HCV infection, anti-HCV therapy, Sjögren’s syndrome, death

## Abstract

Hepatitis C virus (HCV) infection is a potential risk factor for Sjögren’s syndrome (SS). However, it is unclear whether anti-HCV intervention therapy could decrease SS risk. A retrospective cohort analysis from 1997–2012 comprising 17,166 eligible HCV-infected adults was conducted. By 1:2 propensity score matching, a total of 2123 treated patients and 4246 untreated patients were subjected to analysis. The incidence rates and risks of SS and death were evaluated through to the end of 2012. In a total follow-up of 36,906 person-years, 177 (2.8%) patients developed SS, and 522 (8.2%) died during the study period. The incidence rates of SS for the treated and untreated cohorts were 5.3 vs. 4.7/1000 person-years, and those of death for the treated and untreated cohorts were 10.0 vs. 14.8/1000 person-years. A lower risk of death (adjusted hazard ratio, 0.68; 95% CI, 0.53–0.87) was present in HCV-infected patients receiving anti-HCV therapy in multivariable Cox regression, and this remained consistent in multivariable stratified analysis. However, there were no relationships between anti-HCV therapy and its therapeutic duration, and SS risk in multivariable Cox regression. In conclusion, anti-HCV intervention therapy was not associated with lower SS risk in HCV-infected patients, but associated with lower death risk.

## 1. Introduction

Sjögren syndrome (SS) is a systemic autoimmune disease with glandular manifestation (xerophthalmia and xerostomia) and extraglandular manifestation (arthritis, vasculitis, nephritis and interstitial lung disease) [1]. The American–European Consensus Group defined the classification criteria for diagnosing primary SS as evidence of ocular symptoms, oral symptoms, ocular signs (Schirmer’s I test or Rose Bengal score), salivary gland involvement (unstimulated whole salivary flow, sialography, sialoscitigraphy), serological autoimmunity (positive anti-SSA/B autoantibodies), and histopathological evidence (focal lymphocytic sialadenitis with focus score ≥ 1 in a labial salivary gland biopsy) [2,3]. The risk factors for SS development are not yet fully understood. Previous research suggested that genetic precipitation and environmental factors may participate in the development of SS [1].

Viral infection has been proposed as a possible etiologic or triggering agent of autoimmune diseases for decades [4,5,6]. The association between hepatitis C virus (HCV) and SS development was investigated in many previous studies, and is considered to be a potential pathogenic agent in the development of SS [4,7,8,9]. A high prevalence of chronic HCV infection was found in SS patients [4,10,11]. The HCV virus was found in saliva and the salivary gland epithelium in SS [12,13,14]. The HCV envelope protein was reported to recruit lymphocytes to the salivary gland, and induced lymphocytic sialadenitis that was similar to sialadenitis found in SS [15,16,17]. The hepatitis C virus had hepatotropism, lymphotropism, and sialotropism [18]. HCV-infected patients had extrahepatic manifestations such as xerophthalmia and xerostomia, which are similar to SS glandular symptoms, and arthralgia, cryoglobulinemia, and vasculitis, which are similar to SS extraglandular manifestations [4,9,19,20]. A high rate of altered objective tests, such as ocular tests and salivary gland scintigraphy, was also noted in HCV patients [4,9,21]. An increased incidence of abnormal autoantibody production in sera from HCV-infected patients was also found, such as rheumatoid factor, antinuclear, anti-SSA, anti-SSB, and antifodrin antibodies, which frequently appear in SS patients [4,21,22,23,24,25]. Furthermore, a salivary gland biopsy in patients with HCV exocrinopathy is similar to that of chronic sialadenitis in SS [15]. Therefore, HCV infection as a trigger or mimic to SS has always been a controversial position.

Although directly acting antivirals are becoming a new paradigm of HCV therapy [26], anti-HCV intervention therapy with interferon-based therapy (IBT) has been widely used for decades with excellent therapeutic responses in Asian countries, including Taiwan, where an eradication rate of over 70% is prevalent because of the favorable interleukin-28B [27]. Furthermore, detailed information on IBT rather than directly acting antivirals was available for research in the Taiwan’s National Health Insurance (NHI) Research Database (NHIRD) before 2016 because directly acting antivirals have been reimbursed by Taiwan’s single-payer NHI for a minority of HCV-infected patients since 2017, and for all HCV-infected patients after 2019. A small case report revealed that 50% of 12 patients with concurrent SS and HCV receiving IBT had improvement of SS and a sustained virological response [28]. However, the other patients had severe immunological complications. Therefore, it is difficult to tell if the improvement was due to the IBT effect or HCV eradication. The interferon, which is a pleiotropic cytokine, has both immunomodulatory and proinflammatory effects. An exacerbation of disease activity was noted when HCV patients with concurrent rheumatic disease received IBT for HCV infection [29]. Furthermore, previous studies demonstrated that the Type I interferon pathway participated in the pathogenesis of SS [30,31,32]. Although HCV eradication rate by IBT was around 80%, there is a concern whether IBT is safe for HCV patients in the case of future SS development.

Evidence on whether anti-HCV intervention therapy decreases SS risk in HCV-infected patients is scanty. Taiwan is particularly suitable for examining this relationship because it has a higher prevalence of HCV infection. Hence, we examined this association using claims data from the NHIRD.

## 2. Materials and Methods

### 2.1. Data Source

We conducted a retrospective nationwide cohort study that used medical claims data from the NHIRD between 1996 and 2012. The detailed information of NHIRD was described in our previous research [33,34]. In brief, the NHIRD is a nationally representative cohort that is derived from a random sample of all beneficiaries in Taiwan’s compulsory single-payer NHI program and thus contains detailed healthcare information, except for laboratory and lifestyle data. For disease identification, the ICD-9-CM code was used. There was no significant difference in the age distribution, gender distribution, or average insured payroll-related amount between the patients in the NHIRD and NHI programs. The NHIRD is also a deidentified database; thus, this study, with a waiver of informed consent, was exempt from full review by the Institutional Review Board of Dalin Tzu Chi Hospital (B10501028).

### 2.2. Study Population (HCV Cohort)

We first identified 18,418 patients with an ICD-9-CM diagnosis code of HCV (Appendix A) between 1 January 1997 and 31 December 2012 from the outpatient and inpatient claims (Figure 1). We excluded HCV-infected patients who were aged <18 years, had missing data, had a diagnosis of autoimmune rheumatic diseases, HIV infection, and Sjögren’s syndrome before the diagnosis of HCV, and had a diagnosis of Sjögren’s syndrome before anti-HCV therapy, as listed in Appendix A. Lastly, the data of 17,166 adult patients with HCV infection (HCV cohort) were included as the analytic sample.

### 2.3. Anti-HCV Therapy Exposure

The patients were further divided into two groups according to exposure to anti-HCV therapy, namely, interferon-based therapy with interferon used alone or in association with ribavirin [35,36] (Appendix A), which was described in detail in previous studies [35,36]. Directly acting antivirals were not available in the current NHIRD because they have been reimbursed for all HCV-infected patients by Taiwan’s NHI since 2019. The 2126 patients who had experienced anti-HCV therapy were designated as the treated cohort, and 15,040 patients who had never experienced anti-HCV therapy between 1997 and 2012 were designated as the untreated cohort. The treated and untreated cohorts were matched at a 1:2 ratio by propensity scoring to conduct an unbiased estimate of all the confounders predicting anti-HCV therapy prescription. The propensity score was estimated by logistic regression built on the baseline variables of age, sex, comorbidity, geographic region, urbanization level, and number of medical visits. The index date of the treated cohort was the first date of anti-HCV therapy prescription. and that of the untreated cohort was the selected date. The propensity score model was reliable (Hosmer–Lemeshow test, *p* = 0.14) and provided fair discrimination between the cohorts (c-index, 0.61). Lastly, the matched HCV cohort comprised 6369 patients.

### 2.4. Outcome Measures

Our primary study outcome was the occurrence of SS, and the secondary study outcome was total deaths during the follow-up. We did not take into account the cause of death during the follow-up. The identification of SS was obtained from the Registry of Catastrophic Illness Patient Database, a section of the NHIRD. SS is a statutory major disease in Taiwan, and its catastrophic illness certificate is issued after rigorous review of clinical evidence by at least two experienced rheumatologists according to the 1986 American College of Rheumatology Classification criteria and 2002 American–European Consensus Group Classification criteria for Sjögren’s syndrome [3,37], which place the diagnostic accuracy of SS beyond doubt in order to grant exemption from copayment for healthcare. Both cohorts were followed from the index date until the occurrence of SS, death, or the end of 2012, whichever came first.

### 2.5. Covariate Assessment

In addition to age and sex, we considered the comorbidity factor of thyroid disease to minimize the confounding effect of IBT-inducing autoimmune thyroiditis [5,38], the number of medical visits to minimize the detection bias [33,34] and the confounding effect of medical attention [39], geographic regions (northern, central, southern, or eastern Taiwan) to minimize the potential confounding by differential accessibility and availability of urban–rural medical care [33,34,40,41], and urbanization level as a proxy for socioeconomic status to minimize environmental effect [33,34].

### 2.6. Statistical Analysis

Baseline characteristics between the treated and untreated cohorts were compared with a two-sided *t* test for the continuous variables, and χ2 test for the categorical variables. After confirming that there was no violation of the assumption of proportional hazards by plotting the graph of the survival function versus the survival time and the graph of the log (−log(survival)) versus the log of survival time, we applied the Cox proportional hazards regression model to examine the association of anti-HCV therapy with SS risk with adjustment for all covariates (age per year, sex, comorbidity, geographic region, urbanization level, and number of medical visits). We further compared the effect of anti-HCV therapeutic duration (≥3 and ≥6 months) on SS risk, and conducted multivariate stratified analyses in different subgroups to evaluate the effect of anti-HCV therapy on SS and death risks. All data management was performed using SAS (version 9.4; SAS Institute, Inc., Cary, NC, USA) and SPSS (version 20.0; IBM Corp., New York, NY, USA). A two-sided *p*-value less than 0.05 indicated statistical significance.

## 3. Results

### 3.1. Baseline Characteristics

The baseline characteristics of the treated and untreated cohorts are summarized in Table 1. Among the 2126 treated patients, 2123 were matched in a 1:2 ratio to 4246 untreated patients as controls by propensity scores (Figure 1). The proportions of gender, geographic region, urbanization level, the number of medical visits, and propensity score were similar in the two cohorts. Compared with the untreated cohort, the treated cohort was older (average age, 51 years) and had a higher proportion of thyroid disease.

### 3.2. Association between Anti-HCV Therapy for HCV-Infected Patients and Study Outcomes

A total follow-up summation was 36,906 person-years during the study period; 177 patients (2.8%) developed SS, 509 (8%) died before SS development, and total deaths were 522 (8.2%). The numbers of events and incidence rates of SS and death are listed in Table 2. Incidence rates for SS and death were 5.3 and 10.0 events per 1000 person-years in the treated cohort, respectively, vs. 4.7 and 14.8 events per 1000 person-years in the untreated cohort, respectively. After adjusting for confounding covariates, we found that anti-HCV therapy for HCV-infected patients significantly reduced death risk (adjusted hazard ratio (aHR), 0.68; 95% confidence interval (CI), 0.53–0.87) rather than SS risk (0.93; 0.65–1.35). Multivariate Cox proportional hazard analysis (Appendix A) showed that patients who were male (0.34; 0.25–0.48), resided in southern Taiwan (0.58; 0.36–0.92), and suburban (0.65; 0.44–0.95) and rural areas (0.48; 0.30–0.76) exhibited lower SS risk, and those who had thyroid disease (1.61; 1.10–2.34) and more medical visits (1.01; 1.01–1.02), and resided in central Taiwan (2.50; 1.69–3.70) exhibited higher SS risk.

### 3.3. Association between Anti-HCV Therapeutic Duration and SS Risk

The treated cohort who had received less than 6 months (aHR, 0.80; 95% CI, 0.51–1.27) and 6 months or more (aHR, 0.87; 95% CI, 0.42–1.79) of anti-HCV therapy had similarly lower but not significant, risks of SS (Table 3) than those of the untreated cohort.

### 3.4. Multivariable Stratified Analyses

We conducted stratified analyses to test the reliability of our analyses (Figure 2). Multivariate stratified analysis verified that the reduced HRs of death rather than SS in the treated cohort were consistent across all patient subgroups.

## 4. Discussion

To the best of our knowledge, this study is the first large cohort and intervention study to document the long-term impact of anti-HCV therapy for HCV infection on SS risk. After propensity score matching and adjusting for potential confounders, HCV infection treated with anti-HCV therapy was significantly associated with decreased risk of death, which was consistent with previous research [27], rather than SS. Anti-HCV intervention therapy did not alter the risk of SS. Hsu et al. [27] also reported that antiviral treatment in HCV patients did not alter the risk of catastrophic autoimmune diseases. The treatment was unrelated to autoimmune diseases, independent of different strata, including the duration of treatment, age, gender, comorbidity, hospital level, and NSAID use. In our study, we also found no change in SS risk across stratum conditions.

SS is an autoimmune disease with a prevalence of approximate 0.1–0.4% of the general population, with female preponderance, with a female-to-male ratio of about 9:1 [24,42,43], which was consistent with our result in the HCV-infected population. It is also a chronic inflammatory disease involving not only exocrinopathy but also extraglandular systems [1]. It can cause xerophthalmia and xerostomia, and severely impair life quality if left untreated [24]. The exact risk factor for SS development is unknown. Some previous research suggested that genetic precipitation and environment factors such as hormones, infections, and thyroid disease may participate in the development of SS [4,5,38]. Our result also demonstrates thyroid disease to be a risk factor for SS in the HCV-infected population. HCV infection increased the risk of SS development in previous studies [7,8]. Whether IBT in HCV patients could decrease the risk of SS is unknown.

IBT has been widely used for decades for HCV-infected patients and can eradicate viral infection. A sustained virological response can be found in 40% to 80% of patients with HCV [27,44]. Our findings are consistent with those of a previous nationwide cohort study by Cheng et al. [45], who reported that IBT could reduce the overall mortality in HCV-infected patients, but could not reverse HCV-associated risk of rheumatic diseases. A previous case–control study showed a controversial effect of IBT in SS patients with concurrent HCV [28]. Of the patients, 50% had improvement in SS, and 50% had more immunological complications. Interferon has both immunomodulatory and proinflammation effects [29,46]. It can increase the expression of anti-inflammatory molecules such as interleukin-1 receptor antagonist and soluble tumor necrosis factor receptor [47]. However, interferon can also activate dendritic cells, increase MHC class I expression, and may increase the chance of self-antigen presentation and autoreactivity [46,48]. Furthermore, interferon can induce B-cell activating factor (BAFF) production, and then may promote B-cell proliferation and subsequent autoantibody formation [49]. Therefore, the effect of interferon therapy is double-sided. The Type I interferon pathway is a critical factor in the pathogenesis of SS [30,31,32,50]. The polymorphism of interferon regulatory factor 5 and signal transducer and activator of transcription 4 genes, of which the expressed protein participates in the interferon pathway, has a strong association with SS susceptibility [31,51]. Furthermore, the microarray analysis of minor salivary glands, peripheral blood mononuclear cells, and peripheral blood CD14+ monocytes from patients with SS showed that interferon-inducible genes were upregulated, including BAFF [31,52]. An increased expression of BAFF was found in SS [53]. BAFF can promote B-cell activation, survival, and antibody formation. The overexpression of interferon-inducible genes in SS was positively correlated to titers of serum anti-SSA and anti-SSB antibodies [54], which are typical autoantibodies for SS. Minor salivary gland biopsies from patients with SS also showed recruited plasmacytoid dendritic cells, the major source of Type I interferon [55]. Therefore, there was a concern that IBT would cause the development or exacerbation of SS. In this study, we demonstrated that an IBT does not increase the risk of future SS development in viral hepatitis C patients. Our findings indicate that the treatment of HCV infection by IBT may be safe for SS.

The pathogenesis of SS development is unclear. HCV infection has been postulated as a trigger of SS for decades [4,6,7,8]. Viral hepatitis C infection is a chronic viral infection that has a hepatotrophic, lymphotrophic, and sialotropic character [18]. HCV can directly invade the salivary gland epithelium and induce chronic sialadenitis [12,13,14,15,16]. Its envelope protein E2 can promote dendritic cell activation, proinflammatory cytokine production, and B-cell proliferation [16,17]. Elevated BAFF was found in the serum of HCV-infected patients [56]. Therefore, HCV can cause chronic immune activation, lymphoproliferation, and immune complex formation in a host [18]. An abnormal host defense immune response sometimes occurred, and autoimmune diseases developed. The presence of mixed cryoglobulinemia and autoantibodies such as anti-SSA, anti-SS, and antifodrin, which are frequently found in SS, was also noted in HCV-infected patients [4,9,20,21,25]. Hence, this raises the concern of whether antiviral treatment for HCV could theoretically reduce SS risk. However, our results demonstrate that the interferon treatment of HCV does not decrease SS development risk. Previous studies showed that interferon therapy decreased viral loads, even though not curatively, and may halt the autoimmunity process. HCV clearance by IBT may prevent the direct viral invasion of the salivary gland, and may decrease the chronic antiviral immune response, and the chance of self-tolerance breakdown and autoimmune development. However, the benefit of HCV eradication may be neutralized by the proinflammatory effect of IBT treatment [29]. Therefore, in our IBT-treated patients, the risk of SS was not changed.

The study has several strengths. We used a large population-based and highly representative sample with random sampling to reduce selection bias, and considered the use of medical services to reduce detection bias and of urbanization level to reduce environmental effects. In addition, the study population was well-defined, and follow-up was complete because our design relied on computerized registries that provided complete nationwide coverage. Therefore, our finding of unchanged risk of SS in treated HCV-infected patients is robust.

The study also has some potential limitations. First, the adverse reactions and actual compliance related to IBT were not documented from the NHIRD. Nonetheless, excessive prescription is impossible because of the strict regulations for IBT in Taiwan. Second, the misclassification of diseases may occur when an administration database is used. Not all patients with SS seek medical help, and not all clinicians perform a diagnosis of SS and issue a catastrophic-illness certificate. Hence, there was a chance for SS patients to not be included in the Registry of Catastrophic Illness Patient Database. However, the NHI Administration established an audit and penalty system for quality monitoring and assurance to ensure the accuracy of claims. Moreover, the diagnoses of SS and HCV by ICD-9 codes have been applied in several NHIRD-based nationwide cohort studies. Third, the NHIRD lacks information on laboratory data (e.g., sustained virological response, HCV RNA, genotype, and anti-SSA/SSB antibodies), genetic predisposition, lifestyle, and SS severity. Thus, we could not include these variables in the PS analysis, and clarify the relationships of sustained virological response, viral count, genotype, and SS severity. We also could not elucidate the prevalence of true SS with positive anti-SSA/SSB antibodies, resulting in the potential existence of some SS patients who were diagnosed with positive salivary gland biopsy and negative for anti-SSA/SSB antibodies in the current study. Nevertheless, we used propensity score matching to minimize allocation bias in order to reach comparability for the treated and untreated cohorts.

## 5. Conclusions

This national cohort study indicates that IBT fails to decrease SS risk in HCV-infected patients, but decreases death risk. Hence, with regard to the induction of SS development in HCV patients, IBT did not alter the risk. Whether the elimination of the HCV virus by therapies other than IBT (such as directly acting antiviral drugs) can reduce the risk of SS needs further research to help in better understanding the mechanism.

## Figures and Tables

**Figure 1 jcm-11-04259-f001:**
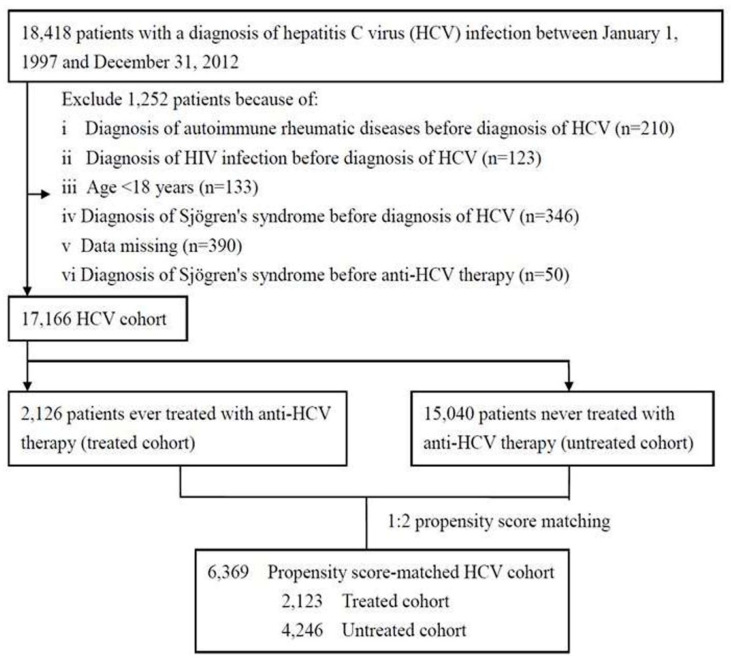
Flow diagram of the enrollment process.

**Figure 2 jcm-11-04259-f002:**
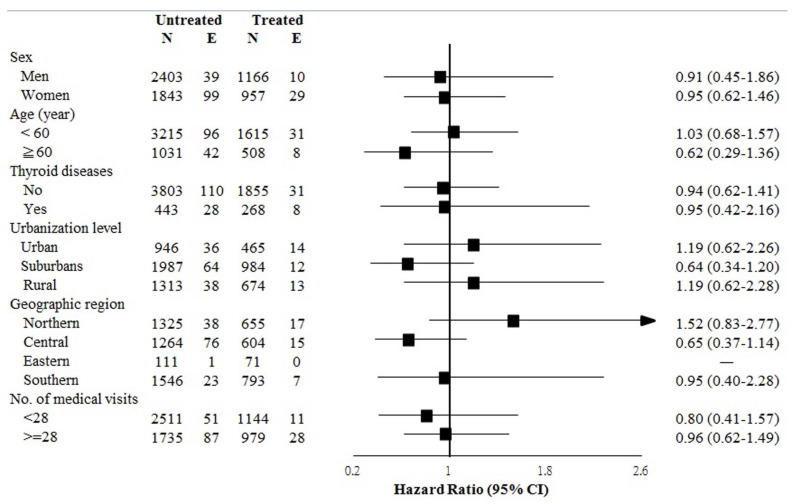
Multivariate stratified analyses for the association between anti-HCV therapy and study outcomes. Each factor was adjusted for all other factors listed in Appendix A. SS, Sjögren’s syndrome; aHR, adjusted hazard ratio; CI, confidence interval.

**Table 1 jcm-11-04259-t001:** Sociodemographic characteristics of the propensity-score-matched HCV cohort in Taiwan in 1997–2012 (*n* = 6369).

Variable	Treated Cohort (*n* = 2123), *n* (%)		Untreated Cohort (*n* = 4246), *n* (%)		*p*-Value *
Sex					0.21
Male	1166	54.9	2403	56.6	
Female	957	45.1	1843	43.4	
Age (year)	51.0 ± 11.6		50.3 ± 13.9		0.024
Comorbidity					
Thyroid disease	268	12.6	443	10.4	0.009
Geographic region					0.28
Northern	655	30.9	1325	31.2	
Central	604	28.5	1264	29.8	
Eastern	71	3.3	111	2.6	
Southern	793	37.4	1546	36.4	
Urbanization level					0.80
Urban	465	21.9	946	22.3	
Suburban	984	46.3	1987	46.8	
Rural	674	31.7	1313	30.9	
Number of medical visits	29.9 ± 21.2		29.0 ± 24.4		0.16
Propensity score	0.14 ± 0.04		0.14 ± 0.04		0.98

Categorical variables given as number (percentage), and continuous variable as mean ± standard deviation. Abbreviation: HCV, hepatitis C virus. * comparison of baseline variables between the treated and untreated cohorts.

**Table 2 jcm-11-04259-t002:** The association of anti-HCV therapy for HCV-infected patients with study outcomes.

			Incidence Rate per 1000 Patient-Years		Study Outcomes, HR (95% CI)			
	No. of Events				SS		Death	
	SS	Death	SS	Death	Crude	Adjusted *	Crude	Adjusted *
Untreated (*n* = 4246)	138 (3.3%)	447 (10.5%)	4.7	14.8	1 (Reference)	1 (Reference)	1 (Reference)	1 (Reference)
Treated (*n* = 2123)	39 (1.8%)	75 (3.5%)	5.3	10.0	0.93 (0.64–1.33)	0.93 (0.65–1.35)	0.68 (0.53–0.87)	0.68 (0.53–0.87)

Abbreviations: SS, Sjögren’s syndrome; HCV, hepatitis C virus; HR, hazard ratio; CI, confidence interval. * Adjusted for age per year, sex, comorbidity, geographic region, urbanization level, and number of medical visits.

**Table 3 jcm-11-04259-t003:** The association between duration of anti-HCV therapy and Sjögren’s syndrome (SS) risk.

	Anti-HCV Therapy Duration	SS Events (%)	Crude HR (95% CI)	*p* Value	Adjusted HR * (95% CI)	*p* Value
Propensity score-matched HCV patients	No (*n* = 4246)	138 (3.3)	1.00 (reference)		1.00 (reference)	
	≧3 ~ <6 months (*n* = 1311)	22 (1.7)	0.79 (0.50–1.25)	0.31	0.80 (0.51–1.27)	0.34
	≧6 months (*n* = 445)	8 (1.8)	0.93 (0.45–1.90)	0.83	0.87 (0.42–1.79)	0.71

Abbreviations: HCV, hepatitis C virus; HR, hazard ratio; CI, confidence interval. * Adjusted for age per year, sex, comorbidity, geographic region, urbanization level, and number of medical visits.

## Data Availability

Restrictions apply to the availability of these data. Data were obtained from the National Health Insurance database and are available from the authors with the permission of the National Health Insurance Administration of Taiwan.

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
