# Peer review of "Association between Anti-Hepatitis C Viral Intervention Therapy and Risk of Sjögren’s Syndrome: A National Retrospective Analysis"

_jcm, 2022, doi:10.3390/jcm11154259_

Round 1

Reviewer 1 Report

The authors have evaluated whether anti-HCV treatment with interferon-based therapeutics can lower the risk of SS development in HCV patients and find that there is no statistically significant benefit for this outcome.  I have a few specific suggestions to improve the text.

1.     On page 1, at the end of the abstract, the authors make the following assertion: “implying time for a reappraisal of the role of HCV infection in the development of SS.” Is this what your findings suggest?  Or is it perhaps that interferon-based treatment for HCV infection in HCV-infected patients does not reduce the risk of SS development? Indeed, there have been some reports and speculation that interferon treatment itself induces SS pathogenesis and interferons are clearly important for SS pathogenesis. Please revise this sentence to reflect what can be concluded from your findings. As you discuss in more detail in the discussion section, the effect of interferon treatment may be reducing HCV infection-based risk of SS while also promoting SS development directly.  Thus, the absence of a reduction in HCV-associated risk following interferon treatment does not negate HCV being a risk factor. 

2.     On page 1 the authors make the following statement: “The risk factor for SS development is unknown.” Age, sex, and other rheumatic diseases (i.e., SLE) have all been identified as risk factors for SS development.  Additionally, prior meta-analysis has reported HCV infection as a risk factor (PMID: 25263827, #8 on reference list) and HCV infection is not the only infection that has been linked to SS as LAMP3 (PMID: 34802379), which is induced by the influenza virus, has been identified as an important contributor to SS pathogenesis.  I realize this manuscript is challenging the validity of HCV infection as a risk factor and the following paragraph discusses the relationship between infection and SS development but starting off this manuscript by saying the risk factors are unknown, is not quite accurate. I’d suggest removing this sentence altogether or changing it to “The risk factors for SS development are not yet fully understood.”  A more nuanced introduction follows already.

3.     On page 2, in the first paragraph, there is a discussion of autoantibodies in HCV-infected patients, but no mention of Anti-SSA/Ro or Anti-SSB/La autoantibodies which are characteristic of SS and are found in HCV-infected patients (PMID: 18692692, 17992468). Please add these autoantibodies to this paragraph.

4.     What do the p-values in Table 1 represent? The legend does not state what analysis was conducted on the demographic information to yield those values, though I presume it is an analysis between the treated and untreated groups. Also, what does “pear year” mean in this table? Details on how the propensity score matching was conducted are not included in the methods. 

5.     This 2021 publication in JCM did a similar study (PMID: 33671397) and also found no effect of interferon treatment for HCV on rheumatic disease development but did find a benefit in overall mortality.  This should be included in your discussion.

6.     In the abstract and introduction, the authors state that they are assessing whether HCV treatment lowers the risk of SS, but then in the conclusion at the end of the paper make this statement: “This national cohort study indicates that IBT fails to increase SS risk in HCV-infected patients, but decrease death risk.”  Did the authors mean fails to decrease SS risk?

7.     What data do you have to support this statement? “Regarding induction of SS development in HCV patients, IBT is safe.”  There are several indications that it exacerbates rheumatic disease.

Author Response

Jul 10, 2022

Clio P. Mavragani, Editor in special issue "Sjogren’s syndrome"

Journal of Clinical Medicine

Re: jcm-1788844

Dear Editors,

Thanks for your letter on Jul 1, 2022 regarding our article titled as: Association Between Anti-Hepatitis C Virus Intervention Therapy and Risk of Sjögren's Syndrome : A National Retrospective Analysis. We've learned a lot from your valuable advice. According to your suggestion, we make some revisions marked as red words in the revised manuscript. More clear details are described as the follows.

Response to Reviewer 1

  1. On page 1, at the end of the abstract, the authors make the following assertion:“implying time for a reappraisal of the role of HCV infection in the development of SS.” Is this what your findings suggest? Or is it perhaps that interferon-based treatment for HCV infection in HCV-infected patients does not reduce the risk of SS development? Indeed, there have been some reports and speculation that interferon treatment itself induces SS pathogenesis and interferons are clearly important for SS pathogenesis. Please revise this sentence to reflect what can be concluded from your findings. As you discuss in more detail in the discussion section, the effect of interferon treatment may be reducing HCV infection-based risk of SS while also promoting SS development directly. Thus, the absence of a reduction in HCV-associated risk following interferon treatment does not negate HCV being a risk factor.

Response: To address the reviewer's concern, we revised our abstract (page 2, lines 14-16).

  1. On page 1 the authors make the following statement: “The risk factor for SS development is unknown.” Age, sex, and other rheumatic diseases (i.e., SLE) have all been identified as risk factors for SS development. Additionally, prior meta-analysis has reported HCV infection as a risk factor (PMID: 25263827, #8 on reference list) and HCV infection is not the only infection that has been linked to SS as LAMP3 (PMID:34802379), which is induced by the influenza virus, has been identified as an important contributor to SS pathogenesis. I realize this manuscript is challenging the validity of HCV infection as a risk factor and the following paragraph discusses the relationship between infection and SS development but starting off this manuscript by saying the risk factors are unknown, is not quite accurate. I’d suggest removing this sentence altogether or changing it to “The risk factors for SS development are not yet fully understood.” A more nuanced introduction follows already.

Response: To address the reviewer's concern, we revised our manuscript (page 3, lines 9-10).

  1. On page 2, in the first paragraph, there is a discussion of autoantibodies in HCV-infected patients, but no mention of Anti-SSA/Ro or Anti-SSB/La autoantibodies which are characteristic of SS and are found in HCV-infected patients (PMID: 18692692,17992468). Please add these autoantibodies to this paragraph.

Response: To address the reviewer's concern, we revised our manuscript (page 4, lines 2-3) and cited two new references (Ref 22,23).

  1. What do the p-values in Table 1 represent? The legend does not state what analysis was conducted on the demographic information to yield those values, though I presume it is an analysis between the treated and untreated groups. Also, what does “pear year” mean in this table? Details on how the propensity score matching was conducted are not included in the methods.

Response: To address the reviewer's concern, we revised our manuscript (page 7, lines 21-22) and Table 1. “Pear year” in table 1 was corrected to "year". The propensity score matching process was included in the methods (page 6, lines 15-17).

  1. This 2021 publication in JCM did a similar study (PMID: 33671397) and also found no effect of interferon treatment for HCV on rheumatic disease development but did find a benefit in overall mortality. This should be included in your discussion.

Response: To address the reviewer's concern, we revised our manuscript (page 10, line 23; page 11, lines 1-2) and cited this reference (Ref 45).

  1. In the abstract and introduction, the authors state that they are assessing whether HCV treatment lowers the risk of SS, but then in the conclusion at the end of the paper make this statement: “This national cohort study indicates that IBT fails to increase SS risk in HCV-infected patients, but decrease death risk.” Did the authors mean fails to decrease SS risk?

Response: To address the reviewer's concern, we revised our manuscript (page 14, line 3).

  1. What data do you have to support this statement? “Regarding induction of SS development in HCV patients, IBT is safe.” There are several indications that it exacerbates rheumatic disease.

Response: To address the reviewer's concern, we revised our manuscript (page 14, lines 4-5).

Thank you heartily for your invaluable opinions on this paper. We are deeply honored by the time and efforts that you had spent in reviewing and revising this manuscript. By incessantly reviewing and revising our texts, we are spurred to read more and learn more from your comments.

Yours sincerely,

Yi-Chun Chen, MD

Division of Nephrology, Department of Internal Medicine, Buddhist Dalin Tzu Chi General Hospital, Chiayi, and School of Medicine, Tzu Chi University, Hualien, Taiwan

Reviewer 2 Report

The retrospective cohort study aimed to clarify the effect of Interferon-based therapy (IBT) on the risk of Sjogren Syndrome (SS) development in HCV-infected patients. The results of the study showed that IBT may significantly decrease the risk of death and has no effect on the risk of SS. The major strengths of the study are the following: big cohort, long period of observation, and proper statistic analysis. In general, the manuscript is clear and presented in a well-structured manner.

From the other side, there are several limitations. First of all, the clinical relevance of the obtained results is doubtful since IBT now is not the method of choice for HCV treatment.  To explain the clinical relevance of your study, add to Introduction section some information related to current use of IBT for HCV infected patients. Moreover, the conclusion statement in the abstract also requires more support as in the study results there is no information if HCV was eradicated in all the treated patients  (“Our finding suggested that anti-HCV intervention therapy was not associated with lower risk of SS, implying time for a reappraisal of the role of HCV infection in the development of SS”).

Also several details should be clarified in the study protocol (Materials and Methods section).

1.     Please give more details related to SS diagnostic criteria used in your study (what diagnostic criteria were accepted in Taiwan in the period of 1997-2012).

2.     Please clarify if all the patients included into the cohort were screened for possible Sjogren Syndrome within the study period. Was there any chance for patients with SS not to be included into the Registry of Catastrophic Illness Patient Database?

3.     Did you assess the prevalence of sicca syndrome without diagnosing of true SS (anti-SSA/SSB antibodies)?

4.     Please give more details related to IBT (was interferon used alone or in association with riboflavin?).

5.     Did you take into account the cause of death? Please clarify if you excluded deaths not associated with HCV.

6.     Please explain the choice of covariates (e.g. Region).

The results of the study are presented clearly and support the main findings.

In the Discussion section please mention that Hsu et al. (25 in the reference list) also assessed the impact of IBT on the autoimmune catastrophic diseases and received similar results. Please pay a little bit more attention to covariates used in the study, discuss the potential impact of each covariate.

The cited literature is mostly relevant despite numbers 31 and 32 in the reference list which seem to be not necessarily.  The majority of listed publications are older than 5 years, but this may be explained by the study period (1997-2012).

Author Response

Jul 10, 2022

Clio P. Mavragani, Editor in special issue "Sjogren’s syndrome"

Journal of Clinical Medicine

Re: jcm-1788844

Dear Editors,

Thanks for your letter on Jul 1, 2022 regarding our article titled as: Association Between Anti-Hepatitis C Virus Intervention Therapy and Risk of Sjögren's Syndrome : A National Retrospective Analysis. We've learned a lot from your valuable advice. According to your suggestion, we make some revisions marked as red words in the revised manuscript. More clear details are described as the follows.

Response to Reviewer 2

  1. Response to "....From the other side, there are several limitations. First of all, the clinical relevance of the obtained results is doubtful since IBT now is not the method of choice for HCV treatment. To explain the clinical relevance of your study, add to Introduction section some information related to current use of IBT for HCV infected patients."

Response: To address the reviewer's concern, we revised our manuscript (page 4, lines 11-13).

  1. Response to "....Moreover, the conclusion statement in the abstract also requires more support as in the study results there is no information if HCV was eradicated in all the treated patients (“Our finding suggested that anti-HCV intervention therapy was not associated with lower risk of SS, implying time for a reappraisal of the role of HCV infection in the development of SS”).

Response: To address the reviewer's concern, we revised our abstract (page 2, lines 14-16).

  1. Response to "1. Please give more details related to SS diagnostic criteria used in your study (what diagnostic criteria were accepted in Taiwan in the period of 1997-2012)."

Response: To address the reviewer's concern, we revised our manuscript (page 7, lines 6-8) and cited a new reference (Ref 37).

  1. Response to "2. Please clarify if all the patients included into the cohort were screened for possible Sjogren Syndrome within the study period. Was there any chance for patients with SS not to be included into the Registry of Catastrophic Illness Patient Database?"

Response: We listed it as a limitation. To address the reviewer's concern, we revised our manuscript (page 13, lines 8-11).

  1. Response to "3. Did you assess the prevalence of sicca syndrome without diagnosing of true SS (anti-SSA/SSBantibodies)?"

Response: We listed it as a limitation. To address the reviewer's concern, we revised our manuscript (page 13, lines 15, 18-21).

  1. Response to "4. Please give more details related to IBT (was interferon used alone or in association with ribavirin?)."

Response: To address the reviewer's concern, we revised our manuscript (page 6, line 7-8). The IBT has been described in detail in previous studies (Ref 35,36).

  1. Response to "5. Did you take into account the cause of death? Please clarify if you excluded deaths not associated with HCV."

Response: We did not take into account the cause of death during the follow-up because we focused on the association between anti-HCV intervention therapy and SS risk as our primary study outcome. Moreover, we took into account the total deaths and did not exclude deaths not associated with HCV. To address the reviewer's concern, we revised our manuscript (page 7, lines 1-3).

  1. Response to "6. Please explain the choice of covariates (e.g. Region)."

Response: To address the reviewer's concern, we revised our manuscript (page 7, lines 16-17) and cited two new references (Ref 40,41).

  1. Response to "In the Discussion section please mention that Hsu et al. (25 in the reference list) also assessed the impact of IBT on the autoimmune catastrophic diseases and received similar results.

Response: To address the reviewer's concern, we revised our manuscript (page 10, lines 4-9).

  1. Response to "Please pay a little bit more attention to covariates used in the study, discuss the potential impact of each covariate."

Response: To address the reviewer's concern, we revised our manuscript (page 7, lines 13-17) and cited some references.

  1. Response to "The cited literature is mostly relevant despite numbers 31 and 32 in the reference list which seem to be not necessarily."

Response: To address the reviewer's concern, we omitted the original Ref 32. The original Ref 31 (now, Ref 34) should be kept and cited for the explanation of covariates (geographic regions of Taiwan, number of medical visits, and urbanization).

Thank you heartily for your invaluable opinions on this paper. We are deeply honored by the time and efforts that you had spent in reviewing and revising this manuscript. By incessantly reviewing and revising our texts, we are spurred to read more and learn more from your comments.

Yours sincerely,

Yi-Chun Chen, MD

Division of Nephrology, Department of Internal Medicine, Buddhist Dalin Tzu Chi General Hospital, Chiayi, and School of Medicine, Tzu Chi University, Hualien, Taiwan

Round 2

Reviewer 2 Report

The authors revised the manusctript and clarified all the details of study design. Also some comments were added regarding the clinical relevance of the obtained results.